# Understanding the Effective Receptive Field in Deep Convolutional Neural Networks

**Wenjie Luo**[*]     **Yujia Li**[*]     **Raquel Urtasun**     **Richard Zemel**
Department of Computer Science
University of Toronto
{wenjie, yujiali, urtasun, zemel}@cs.toronto.edu

## Abstract

We study characteristics of receptive fields of units in deep convolutional networks. The receptive field size is a crucial issue in many visual tasks, as the output must respond to large enough areas in the image to capture information about large objects. We introduce the notion of an effective receptive field, and show that it both has a Gaussian distribution and only occupies a fraction of the full theoretical receptive field. We analyze the effective receptive field in several architecture designs, and the effect of nonlinear activations, dropout, sub-sampling and skip connections on it. This leads to suggestions for ways to address its tendency to be too small.

## 1  Introduction

Deep convolutional neural networks (CNNs) have achieved great success in a wide range of problems in the last few years. In this paper we focus on their application to computer vision: where they are the driving force behind the significant improvement of the state-of-the-art for many tasks recently, including image recognition [10, 8], object detection [17, 2], semantic segmentation [12, 1], image captioning [20], and many more.

One of the basic concepts in deep CNNs is the *receptive field*, or *field of view*, of a unit in a certain layer in the network. Unlike in fully connected networks, where the value of each unit depends on the entire input to the network, a unit in convolutional networks only depends on a region of the input. This region in the input is the receptive field for that unit.

The concept of receptive field is important for understanding and diagnosing how deep CNNs work. Since anywhere in an input image outside the receptive field of a unit does not affect the value of that unit, it is necessary to carefully control the receptive field, to ensure that it covers the entire relevant image region. In many tasks, especially dense prediction tasks like semantic image segmentation, stereo and optical flow estimation, where we make a prediction for each single pixel in the input image, it is critical for each output pixel to have a big receptive field, such that no important information is left out when making the prediction.

The receptive field size of a unit can be increased in a number of ways. One option is to stack more layers to make the network deeper, which increases the receptive field size linearly by theory, as each extra layer increases the receptive field size by the kernel size. Sub-sampling on the other hand increases the receptive field size multiplicatively. Modern deep CNN architectures like the VGG networks [18] and Residual Networks [8, 6] use a combination of these techniques.

In this paper, we carefully study the receptive field of deep CNNs, focusing on problems in which there are many output unites. In particular, we discover that not all pixels in a receptive field contribute equally to an output unit's response. Intuitively it is easy to see that pixels at the center of a receptive

---

[*]denotes equal contribution

field have a much larger impact on an output. In the forward pass, central pixels can propagate information to the output through many different paths, while the pixels in the outer area of the receptive field have very few paths to propagate its impact. In the backward pass, gradients from an output unit are propagated across all the paths, and therefore the central pixels have a much larger magnitude for the gradient from that output.

This observation leads us to study further the distribution of impact within a receptive field on the output. Surprisingly, we can prove that in many cases the distribution of impact in a receptive field distributes as a Gaussian. Note that in earlier work [20] this Gaussian assumption about a receptive field is used without justification. This result further leads to some intriguing findings, in particular that the effective area in the receptive field, which we call the *effective receptive field*, only occupies a fraction of the *theoretical receptive field*, since Gaussian distributions generally decay quickly from the center.

The theory we develop for effective receptive field also correlates well with some empirical observations. One such empirical observation is that the currently commonly used random initializations lead some deep CNNs to start with a small effective receptive field, which then grows during training. This potentially indicates a bad initialization bias.

Below we present the theory in Section 2 and some empirical observations in Section 3, which aim at understanding the effective receptive field for deep CNNs. We discuss a few potential ways to increase the effective receptive field size in Section 4.

## 2    Properties of Effective Receptive Fields

We want to mathematically characterize how much each input pixel in a receptive field can impact the output of a unit $n$ layers up the network, and study how the impact distributes within the receptive field of that output unit. To simplify notation we consider only a single channel on each layer, but similar results can be easily derived for convolutional layers with more input and output channels.

Assume the pixels on each layer are indexed by $(i, j)$, with their center at $(0, 0)$. Denote the $(i, j)$th pixel on the $p$th layer as $x_{i,j}^p$, with $x_{i,j}^0$ as the input to the network, and $y_{i,j} = x_{i,j}^n$ as the output on the $n$th layer. We want to measure how much each $x_{i,j}^0$ contributes to $y_{0,0}$. We define the *effective receptive field* (ERF) of this central output unit as region containing any input pixel with a non-negligible impact on that unit.

The measure of impact we use in this paper is the partial derivative $\partial y_{0,0}/\partial x_{i,j}^0$. It measures how much $y_{0,0}$ changes as $x_{i,j}^0$ changes by a small amount; it is therefore a natural measure of the importance of $x_{i,j}^0$ with respect to $y_{0,0}$. However, this measure depends not only on the weights of the network, but are in most cases also input-dependent, so most of our results will be presented in terms of expectations over input distribution.

The partial derivative $\partial y_{0,0}/\partial x_{i,j}^0$ can be computed with back-propagation. In the standard setting, back-propagation propagates the error gradient with respect to a certain loss function. Assuming we have an arbitrary loss $l$, by the chain rule we have $\frac{\partial l}{\partial x_{i,j}^0} = \sum_{i',j'} \frac{\partial l}{\partial y_{i',j'}} \frac{\partial y_{i',j'}}{\partial x_{i,j}^0}$.

Then to get the quantity $\partial y_{0,0}/\partial x_{i,j}^0$, we can set the error gradient $\partial l/\partial y_{0,0} = 1$ and $\partial l/\partial y_{i,j} = 0$ for all $i \neq 0$ and $j \neq 0$, then propagate this gradient from there back down the network. The resulting $\partial l/\partial x_{i,j}^0$ equals the desired $\partial y_{0,0}/\partial x_{i,j}^0$. Here we use the back-propagation process without an explicit loss function, and the process can be easily implemented with standard neural network tools.

In the following we first consider linear networks, where this derivative does not depend on the input and is purely a function of the network weights and $(i, j)$, which clearly shows how the impact of the pixels in the receptive field distributes. Then we move forward to consider more modern architecture designs and discuss the effect of nonlinear activations, dropout, sub-sampling, dilation convolution and skip connections on the ERF.

### 2.1    The simplest case: a stack of convolutional layers of weights all equal to one

Consider the case of $n$ convolutional layers using $k \times k$ kernels with stride one, one single channel on each layer and no nonlinearity, stacked into a deep linear CNN. In this analysis we ignore the biases on all layers. We begin by analyzing convolution kernels with weights all equal to one.

Denote $g(i,j,p) = \partial l / \partial x_{i,j}^p$ as the gradient on the $p$th layer, and let $g(i,j,n) = \partial l / \partial y_{i,j}$. Then $g(,,0)$ is the desired gradient image of the input. The back-propagation process effectively convolves $g(,,p)$ with the $k \times k$ kernel to get $g(,,p-1)$ for each $p$.

In this special case, the kernel is a $k \times k$ matrix of 1's, so the 2D convolution can be decomposed into the product of two 1D convolutions. We therefore focus exclusively on the 1D case. We have the initial gradient signal $u(t)$ and kernel $v(t)$ formally defined as

$$u(t) = \delta(t), \quad v(t) = \sum_{m=0}^{k-1} \delta(t - m), \quad \text{where } \delta(t) = \left\{ \begin{array}{ll} 1, & t = 0 \\ 0, & t \neq 0 \end{array} \right. \tag{1}$$

and $t = 0, 1, -1, 2, -2, ...$ indexes the pixels.

The gradient signal on the input pixels is simply $o = u * v * \cdots * v$, convolving $u$ with $n$ such $v$'s. To compute this convolution, we can use the Discrete Time Fourier Transform to convert the signals into the Fourier domain, and obtain

$$U(\omega) = \sum_{t=-\infty}^{\infty} u(t)e^{-j\omega t} = 1, \quad V(\omega) = \sum_{t=-\infty}^{\infty} v(t)e^{-j\omega t} = \sum_{m=0}^{k-1} e^{-j\omega m} \tag{2}$$

Applying the convolution theorem, we have the Fourier transform of $o$ is

$$\mathcal{F}(o) = \mathcal{F}(u * v * \cdots * v)(\omega) = U(\omega) \cdot V(\omega)^n = \left( \sum_{m=0}^{k-1} e^{-j\omega m} \right)^n \tag{3}$$

Next, we need to apply the inverse Fourier transform to get back $o(t)$:

$$o(t) = \frac{1}{2\pi} \int_{-\pi}^{\pi} \left( \sum_{m=0}^{k-1} e^{-j\omega m} \right)^n e^{j\omega t} \mathrm{d}\omega \tag{4}$$

$$\frac{1}{2\pi} \int_{-\pi}^{\pi} e^{-j\omega s} e^{j\omega t} \mathrm{d}\omega = \left\{ \begin{array}{ll} 1, & s = t \\ 0, & s \neq t \end{array} \right. \tag{5}$$

We can see that $o(t)$ is simply the coefficient of $e^{-j\omega t}$ in the expansion of $\left( \sum_{m=0}^{k-1} e^{-j\omega m} \right)^n$.

**Case $k = 2$:** Now let's consider the simplest nontrivial case of $k = 2$, where $\left( \sum_{m=0}^{k-1} e^{-j\omega m} \right)^n = (1 + e^{-j\omega})^n$. The coefficient for $e^{-j\omega t}$ is then the standard binomial coefficient $\binom{n}{t}$, so $o(t) = \binom{n}{t}$. It is quite well known that binomial coefficients distributes with respect to $t$ like a Gaussian as $n$ becomes large (see for example [13]), which means the scale of the coefficients decays as a squared exponential as $t$ deviates from the center. When multiplying two 1D Gaussian together, we get a 2D Gaussian, therefore in this case, the gradient on the input plane is distributed like a 2D Gaussian.

**Case $k > 2$:** In this case the coefficients are known as "extended binomial coefficients" or "polynomial coefficients", and they too distribute like Gaussian, see for example [3, 16]. This is included as a special case for the more general case presented later in Section 2.3.

## 2.2 Random weights

Now let's consider the case of random weights. In general, we have

$$g(i,j,p-1) = \sum_{a=0}^{k-1} \sum_{b=0}^{k-1} w_{a,b}^p g(i+a, i+b, p) \tag{6}$$

with pixel indices properly shifted for clarity, and $w_{a,b}^p$ is the convolution weight at $(a,b)$ in the convolution kernel on layer $p$. At each layer, the initial weights are independently drawn from a fixed distribution with zero mean and variance $C$. We assume that the gradients $g$ are independent from the weights. This assumption is in general not true if the network contains nonlinearities, but for linear networks these assumptions hold. As $\mathbb{E}_w[w_{a,b}^p] = 0$, we can then compute the expectation

$$\mathbb{E}_{w,input}[g(i,j,p-1)] = \sum_{a=0}^{k-1} \sum_{b=0}^{k-1} \mathbb{E}_w[w_{a,b}^p] \mathbb{E}_{input}[g(i+a, i+b, p)] = 0, \quad \forall p \tag{7}$$

Here the expectation is taken over $w$ distribution as well as the input data distribution. The variance is more interesting, as

$$\text{Var}[g(i,j,p-1)] = \sum_{a=0}^{k-1}\sum_{b=0}^{k-1} \text{Var}[w_{a,b}^p]\text{Var}[g(i+a,i+b,p)] = C\sum_{a=0}^{k-1}\sum_{b=0}^{k-1} \text{Var}[g(i+a,i+b,p)] \quad (8)$$

This is equivalent to convolving the gradient variance image $\text{Var}[g(,,p)]$ with a $k \times k$ convolution kernel full of 1's, and then multiplying by $C$ to get $\text{Var}[g(,,p-1)]$.

Based on this we can apply exactly the same analysis as in Section 2.1 on the gradient variance images. The conclusions carry over easily that $\text{Var}[g(.,.,0)]$ has a Gaussian shape, with only a slight change of having an extra $C^n$ constant factor multiplier on the variance gradient images, which does not affect the relative distribution within a receptive field.

### 2.3  Non-uniform kernels

More generally, each pixel in the kernel window can have different weights, or as in the random weight case, they may have different variances. Let's again consider the 1D case, $u(t) = \delta(t)$ as before, and the kernel signal $v(t) = \sum_{m=0}^{k-1} w(m)\delta(t-m)$, where $w(m)$ is the weight for the $m$th pixel in the kernel. Without loss of generality, we can assume the weights are normalized, i.e. $\sum_m w(m) = 1$.

Applying the Fourier transform and convolution theorem as before, we get

$$U(\omega) \cdot V(\omega) \cdots V(\omega) = \left(\sum_{m=0}^{k-1} w(m)e^{-j\omega m}\right)^n \quad (9)$$

the space domain signal $o(t)$ is again the coefficient of $e^{-j\omega t}$ in the expansion; the only difference is that the $e^{-j\omega m}$ terms are weighted by $w(m)$.

These coefficients turn out to be well studied in the combinatorics literature, see for example [3] and the references therein for more details. In [3], it was shown that if $w(m)$ are normalized, then $o(t)$ exactly equals to the probability $p(S_n = t)$, where $S_n = \sum_{i=1}^n X_i$ and $X_i$'s are i.i.d. multinomial variables distributed according to $w(m)$'s, i.e. $p(X_i = m) = w(m)$. Notice the analysis there requires that $w(m) > 0$. But we can reduce to variance analysis for the random weight case, where the variances are always nonnegative while the weights can be negative. The analysis for negative $w(m)$ is more difficult and is left to future work. However empirically we found the implications of the analysis in this section still applies reasonably well to networks with negative weights.

From the central limit theorem point of view, as $n \to \infty$, the distribution of $\sqrt{n}(\frac{1}{n}S_n - \mathbb{E}[X])$ converges to Gaussian $\mathcal{N}(0, \text{Var}[X])$ in distribution. This means, for a given $n$ large enough, $S_n$ is going to be roughly Gaussian with mean $n\mathbb{E}[X]$ and variance $n\text{Var}[X]$. As $o(t) = p(S_n = t)$, this further implies that $o(t)$ also has a Gaussian shape. When $w(m)$'s are normalized, this Gaussian has the following mean and variance:

$$\mathbb{E}[S_n] = n\sum_{m=0}^{k-1} mw(m), \quad \text{Var}[S_n] = n\left(\sum_{m=0}^{k-1} m^2 w(m) - \left(\sum_{m=0}^{k-1} mw(m)\right)^2\right) \quad (10)$$

This indicates that $o(t)$ decays from the center of the receptive field squared exponentially according to the Gaussian distribution. The rate of decay is related to the variance of this Gaussian. If we take one standard deviation as the *effective receptive field (ERF) size* which is roughly the radius of the ERF, then this size is $\sqrt{\text{Var}[S_n]} = \sqrt{n\text{Var}[X_i]} = O(\sqrt{n})$.

On the other hand, as we stack more convolutional layers, the theoretical receptive field grows linearly, therefore relative to the theoretical receptive field, the ERF actually *shrinks* at a rate of $O(1/\sqrt{n})$, which we found surprising.

In the simple case of uniform weighting, we can further see that the ERF size grows linearly with kernel size $k$. As $w(m) = 1/k$, we have

$$\sqrt{\text{Var}[S_n]} = \sqrt{n}\sqrt{\sum_{m=0}^{k-1}\frac{m^2}{k} - \left(\sum_{m=0}^{k-1}\frac{m}{k}\right)^2} = \sqrt{\frac{n(k^2-1)}{12}} = O(k\sqrt{n}) \quad (11)$$

**Remarks:** The result derived in this section, i.e., the distribution of impact within a receptive field in deep CNNs converges to Gaussian, holds under the following conditions: (1) all layers in the CNN use the same set of convolution weights. This is in general not true, however, when we apply the analysis of variance, the weight variance on all layers are usually the same up to a constant factor. (2) The convergence derived is convergence "in distribution", as implied by the central limit theorem. This means that the cumulative probability distribution function converges to that of a Gaussian, but at any single point in space the probability can deviate from the Gaussian. (3) The convergence result states that $\sqrt{n}(\frac{1}{n}S_n - \mathbb{E}[X]) \rightarrow \mathcal{N}(0, \text{Var}[X])$, hence $S_n$ approaches $\mathcal{N}(n\mathbb{E}[X], n\text{Var}[X])$, however the convergence of $S_n$ here is not well defined as $\mathcal{N}(n\mathbb{E}[X], n\text{Var}[X])$ is not a fixed distribution, but instead it changes with $n$. Additionally, the distribution of $S_n$ can deviate from Gaussian on a finite set. But the overall shape of the distribution is still roughly Gaussian.

## 2.4 Nonlinear activation functions

Nonlinear activation functions are an integral part of every neural network. We use $\sigma$ to represent an arbitrary nonlinear activation function. During the forward pass, on each layer the pixels are first passed through $\sigma$ and then convolved with the convolution kernel to compute the next layer. This ordering of operations is a little non-standard but equivalent to the more usual ordering of convolving first and passing through nonlinearity, and it makes the analysis slightly easier. The backward pass in this case becomes

$$g(i,j,p-1) = {\sigma_{i,j}^p}' \sum_{a=0}^{k-1} \sum_{b=0}^{k-1} w_{a,b}^p g(i+a, i+b, p) \tag{12}$$

where we abused notation a bit and use ${\sigma_{i,j}^p}'$ to represent the gradient of the activation function for pixel $(i,j)$ on layer $p$.

For ReLU nonlinearities, ${\sigma_{i,j}^p}' = \mathbf{I}[x_{i,j}^p > 0]$ where $\mathbf{I}[.]$ is the indicator function. We have to make some extra assumptions about the activations $x_{i,j}^p$ to advance the analysis, in addition to the assumption that it has zero mean and unit variance. A standard assumption is that $x_{i,j}^p$ has a symmetric distribution around 0 [7]. If we make an extra simplifying assumption that the gradients $\sigma'$ are independent from the weights and $g$ in the upper layers, we can simplify the variance as $\text{Var}[g(i,j,p-1)] = \mathbb{E}[{\sigma_{i,j}^p}'^2] \sum_a \sum_b \text{Var}[w_{a,b}^p]\text{Var}[g(i+a, i+b, p)]$, and $\mathbb{E}[{\sigma_{i,j}^p}'^2] = \text{Var}[{\sigma_{i,j}^p}'] = 1/4$ is a constant factor. Following the variance analysis we can again reduce this case to the uniform weight case.

Sigmoid and Tanh nonlinearities are harder to analyze. Here we only use the observation that when the network is initialized the weights are usually small and therefore these nonlinearities will be in the linear region, and the linear analysis applies. However, as the weights grow bigger during training their effect becomes hard to analyze.

## 2.5 Dropout, Subsampling, Dilated Convolution and Skip-Connections

Here we consider the effect of some standard CNN approaches on the effective receptive field. Dropout is a popular technique to prevent overfitting; we show that dropout does not change the Gaussian ERF shape. Subsampling and dilated convolutions turn out to be effective ways to increase receptive field size quickly. Skip-connections on the other hand make ERFs smaller. We present the analysis for all these cases in the Appendix.

# 3 Experiments

In this section, we empirically study the ERF for various deep CNN architectures. We first use artificially constructed CNN models to verify the theoretical results in our analysis. We then present our observations on how the ERF changes during the training of deep CNNs on real datasets. For all ERF studies, we place a gradient signal of 1 at the center of the output plane and 0 everywhere else, and then back-propagate this gradient through the network to get input gradients.

## 3.1 Verifying theoretical results

We first verify our theoretical results in artificially constructed deep CNNs. For computing the ERF we use random inputs, and for all the random weight networks we followed [7, 5] for proper random initialization. In this section, we verify the following results:

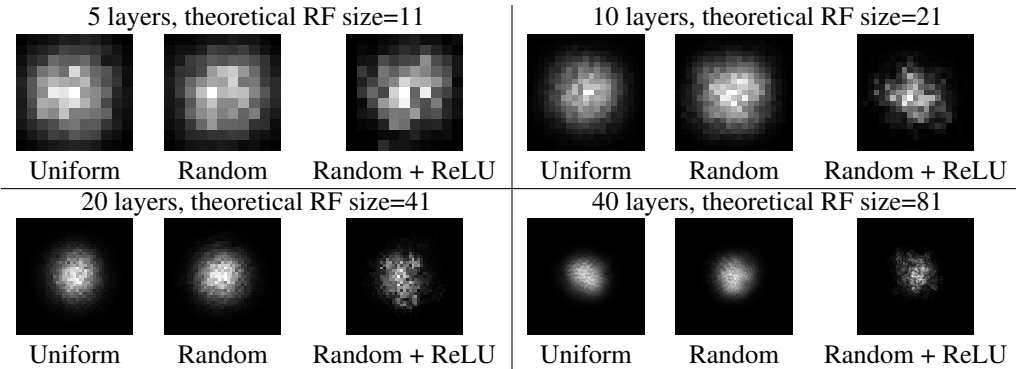

Figure 1: Comparing the effect of number of layers, random weight initialization and nonlinear activation on the ERF. Kernel size is fixed at $3 \times 3$ for all the networks here. Uniform: convolutional kernel weights are all ones, no nonlinearity; Random: random kernel weights, no nonlinearity; Random + ReLU: random kernel weights, ReLU nonlinearity.

**ERFs are Gaussian distributed:** As shown in Fig. 1, we can observe perfect Gaussian shapes for uniformly and randomly weighted convolution kernels without nonlinear activations, and near Gaussian shapes for randomly weighted kernels with nonlinearity. Adding the ReLU nonlinearity makes the distribution a bit less Gaussian, as the ERF distribution depends on the input as well. Another reason is that ReLU units output exactly zero for half of its inputs and it is very easy to get a zero output for the center pixel on the output plane, which means no path from the receptive field can reach

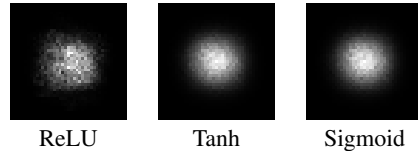

the output, hence the gradient is all zero. Here the ERFs are averaged over 20 runs with different random seed. The figures on the right shows the ERF for networks with 20 layers of random weights, with different nonlinearities. Here the results are averaged both across 100 runs with different random weights as well as different random inputs. In this setting the receptive fields are a lot more Gaussian-like.

$\sqrt{n}$ **absolute growth and** $1/\sqrt{n}$ **relative shrinkage:** In Fig. 2, we show the change of ERF size and the relative ratio of ERF over theoretical RF w.r.t number of convolution layers. The best fitting line for ERF size gives slope of 0.56 in log domain, while the line for ERF ratio gives slope of -0.43. This indicates ERF size is growing linearly w.r.t $\sqrt{N}$ and ERF ratio is shrinking linearly w.r.t. $\frac{1}{\sqrt{N}}$. Note here we use 2 standard deviations as our measurement for ERF size, i.e. any pixel with value greater than $1 - 95.45\%$ of center point is considered as in ERF. The ERF size is represented by the square root of number of pixels within ERF, while the theoretical RF size is the side length of the square in which all pixel has a non-zero impact on the output pixel, no matter how small. All experiments here are averaged over 20 runs.

**Subsampling & dilated convolution increases receptive field:** The figure on the right shows the effect of subsampling and dilated convolution. The reference baseline is a convnet with 15 dense convolution layers. Its ERF is shown in the left-most figure. We then replace 3 of the 15 convolutional layers with stride-2 convolution to get the ERF for the 'Subsample' figure, and replace them with dilated convolution with factor 2,4 and 8 for the 'Dilation' figure. As we see, both of them are able to increase the effect receptive field significantly. Note the 'Dilation' figure shows a rectangular ERF shape typical for dilated convolutions.

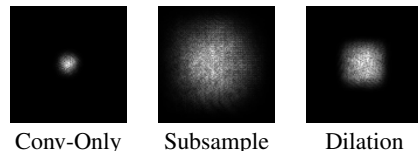

## 3.2 How the ERF evolves during training

In this part, we take a look at how the ERF of units in the top-most convolutional layers of a classification CNN and a semantic segmentation CNN evolve during training. For both tasks, we adopt the ResNet architecture which makes extensive use of skip-connections. As the analysis shows, the ERF of this network should be significantly smaller than the theoretical receptive field. This is indeed what we have observed initially. Intriguingly, as the networks learns, the ERF gets bigger, and at the end of training is significantly larger than the initial ERF.

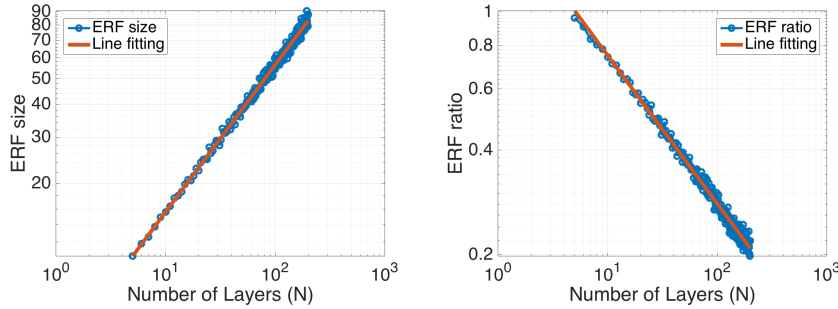

Figure 2: Absolute growth (left) and relative shrink (right) for ERF

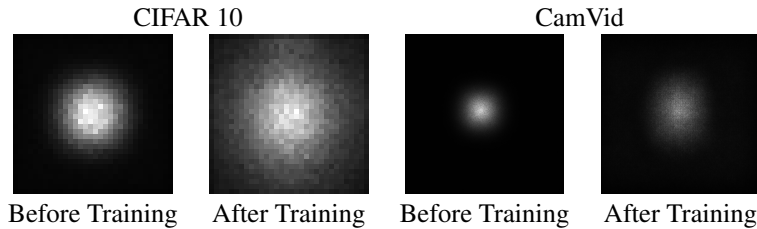

| CIFAR 10 | | CamVid | |
| Before Training | After Training | Before Training | After Training |

Figure 3: Comparison of ERF before and after training for models trained on CIFAR-10 classification and CamVid semantic segmentation tasks. CIFAR-10 receptive fields are visualized in the image space of $32 \times 32$.

For the classification task we trained a ResNet with 17 residual blocks on the CIFAR-10 dataset. At the end of training this network reached a test accuracy of 89%. Note that in this experiment we did not use pooling or downsampling, and exclusively focus on architectures with skip-connections. The accuracy of the network is not state-of-the-art but still quite high. In Fig. 3 we show the effective receptive field on the $32 \times 32$ image space at the beginning of training (with randomly initialized weights) and at the end of training when it reaches best validation accuracy. Note that the theoretical receptive field of our network is actually $74 \times 74$, bigger than the image size, but the ERF is still not able to fully fill the image. Comparing the results before and after training, we see that the effective receptive field has grown significantly.

For the semantic segmentation task we used the CamVid dataset for urban scene segmentation. We trained a "front-end" model [21] which is a purely convolutional network that predicts the output at a slightly lower resolution. This network plays the same role as the VGG network does in many previous works [12]. We trained a ResNet with 16 residual blocks interleaved with 4 subsampling operations each with a factor of 2. Due to these subsampling operations the output is $1/16$ of the input size. For this model, the theoretical receptive field of the top convolutional layer units is quite big at $505 \times 505$. However, as shown in Fig. 3, the ERF only gets a fraction of that with a diameter of 100 at the beginning of training. Again we observe that during training the ERF size increases and at the end it reaches almost a diameter around 150.

## 4 Reduce the Gaussian Damage

The above analysis shows that the ERF only takes a small portion of the theoretical receptive field, which is undesirable for tasks that require a large receptive field.

**New Initialization.** One simple way to increase the effective receptive field is to manipulate the initial weights. We propose a new random weight initialization scheme that makes the weights at the center of the convolution kernel to have a smaller scale, and the weights on the outside to be larger; this diffuses the concentration on the center out to the periphery. Practically, we can initialize the network with any initialization method, then scale the weights according to a distribution that has a lower scale at the center and higher scale on the outside.

In the extreme case, we can optimize the $w(m)$'s to maximize the ERF size or equivalently the variance in Eq. 10. Solving this optimization problem leads to the solution that put weights equally at the 4 corners of the convolution kernel while leaving everywhere else 0. However, using this solution to do random weight initialization is too aggressive, and leaving a lot of weights to 0 makes learning slow. A softer version of this idea usually works better.

We have trained a CNN for the CIFAR-10 classification task with this initialization method, with several random seeds. In a few cases we get a 30% speed-up of training compared to the more standard initializations [5, 7]. But overall the benefit of this method is not always significant.

We note that no matter what we do to change $w(m)$, the effective receptive field is still distributed like a Gaussian so the above proposal only solves the problem partially.

**Architectural changes.** A potentially better approach is to make architectural changes to the CNNs, which may change the ERF in more fundamental ways. For example, instead of connecting each unit in a CNN to a local rectangular convolution window, we can sparsely connect each unit to a larger area in the lower layer using the same number of connections. Dilated convolution [21] belongs to this category, but we may push even further and use sparse connections that are not grid-like.

## 5 Discussion

**Connection to biological neural networks.** In our analysis we have established that the effective receptive field in deep CNNs actually grows a lot slower than we used to think. This indicates that a lot of local information is still preserved even after many convolution layers. This finding contradicts some long-held relevant notions in deep biological networks. A popular characterization of mammalian visual systems involves a split into "what" and "where" pathways [19]. Progressing along the what or where pathway, there is a gradual shift in the nature of connectivity: receptive field sizes increase, and spatial organization becomes looser until there is no obvious retinotopic organization; the loss of retinotopy means that single neurons respond to objects such as faces anywhere in the visual field [9]. However, if the ERF is smaller than the RF, this suggests that representations may retain position information, and also raises an interesting question concerning changes in the size of these fields during development.

A second relevant effect of our analysis is that it suggests that convolutional networks may automatically create a form of foveal representation. The fovea of the human retina extracts high-resolution information from an image only in the neighborhood of the central pixel. Sub-fields of equal resolution are arranged such that their size increases with the distance from the center of the fixation. At the periphery of the retina, lower-resolution information is extracted, from larger regions of the image. Some neural networks have explicitly constructed representations of this form [11]. However, because convolutional networks form Gaussian receptive fields, the underlying representations will naturally have this character.

**Connection to previous work on CNNs.** While receptive fields in CNNs have not been studied extensively, [7, 5] conduct similar analyses, in terms of computing how the variance evolves through the networks. They developed a good initialization scheme for convolution layers following the principle that variance should not change much when going through the network.

Researchers have also utilized visualizations in order to understand how neural networks work. [14] showed the importance of using natural-image priors and also what an activation of the convolutional layer would represent. [22] used deconvolutional nets to show the relation of pixels in the image and the neurons that are firing. [23] did empirical study involving receptive field and used it as a cue for localization. There are also visualization studies using gradient ascent techniques [4] that generate interesting images, such as [15]. These all focus on the unit activations, or feature map, instead of the effective receptive field which we investigate here.

## 6 Conclusion

In this paper, we carefully studied the receptive fields in deep CNNs, and established a few surprising results about the effective receptive field size. In particular, we have shown that the distribution of impact within the receptive field is asymptotically Gaussian, and the effective receptive field only takes up a fraction of the full theoretical receptive field. Empirical results echoed the theory we established. We believe this is just the start of the study of effective receptive field, which provides a new angle to understand deep CNNs. In the future we hope to study more about what factors impact effective receptive field in practice and how we can gain more control over them.

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
