[Reviews · NeurIPS 2016]

Reviewer 1

Summary

The paper introduces the notion of effective receptive-field size in convnets, showing that it grows far slower with depth than the theoretical receptive field size growth one usually thinks of.

Qualitative Assessment

I believe the introduction of the notion of effective receptive field size and its detailed analysis is novel and something the community is unaware of and would benefit knowing about. The paper is overall well written, and conveys some possibly useful insights. It would have been much more valuable however if the authors had been able to leverage these insights in some way to show improvement upon current deep network initialization or training procedures (e.g. obtaining similar or better performance out of a smaller net with e.g. filter initialization or elementwise learning rates derived from these insights). - When you show effective receptive fields of random weight nets, is what you are showing the variances of the gradients?? (as you earlier stated that their expectation would be 0). Please state explicitly what it is your figures are showing. - You say that you used random inputs in your experiments. Did you also try with real natural image inputs, in particular for those networks trained on them? It would be interesting to see if it changes anything. More detailed comments and typos l 28: "deepershow" -> "deeper" In Equations 7,8,9: I believe it should be "j+b" instead of "i+b" l 98: "independent of the gradients" comment in a footnote on the validity of that assumption l 105: "The conclusions carry over easily" -> restate/adapt specifically the carried over conclusions l 111: "only consier w(m)>0": why? and to what degree does it limit the analysis l 160: "many more layers are needed to see the Gaussian shape clearly" -> be more specific, or give/show an example l 164: "[]" missing reference l 169,170,174 "a effective" -> "an effective" l 170: "that's Gaussian" -> "that is Gaussian" Figure 1 caption: "are all the ones" -> "are all ones" l 278: "slower than we use to think" -> "slower than we used to think" l 281: "the where or object" -> didn't you mean the "what"?? l 293,294 "because ... will naturally ..." this is not so obvious, please elaborate.

Confidence in this Review

2-Confident (read it all; understood it all reasonably well)


Reviewer 2

Summary

This paper shows using theoretical arguments that the effective receptive field size in a deep convolutional network is Gaussian shaped (as the number of layers -> infinity) and more specifically that its size shrinks relative to the size of the “theoretical receptive field size” of a unit. The authors suggest that initializations that start with a small effective receptive field can reflect a bad learning bias.

Qualitative Assessment

Overall I found the main result of this paper to be quite interesting since it gives us a theoretical way to think about how to keep receptive field sizes “large enough”. But I found the theoretical section to be a bit convoluted. From my perspective, it would have been easiest to skip to the general case (Section 2.3) in which the central limit theorem is invoked. I didn’t find the discussion of the special cases before this part to be particularly illuminating, and unless I am mistaken, they are completely subsumed by Section 2.3. Beyond this, the exposition in the paper is understandable enough, but could still be improved significantly. I recommend a careful proofreading. Some of the theoretical results are somewhat buried in the discussion, and it would be clearer to lift them out and present them as propositions/lemmas/theorems etc so that the assumptions and conclusions are separated from the justification. Some questions I have about the analysis: * The authors do not seem to discuss the multiple channel case of the problem, and it’s not obvious to me whether the same theoretical arguments would hold for this setting. * I did not fully understand the discussion about adding average pooling with skip connections -> perhaps a diagram would help here. * The abstract promises a discussion of the impact of dropout, which I did not find in the main text. Another thing that is common in typical convolutional networks these days is batch normalization, which is also not discussed, but might help justify the extra assumptions needed to show that the main results hold using ReLU nonlinearities. In general I thought that the experiments could have been carried out in a more quantitative way. It’s somewhat interesting that the receptive field is able to grow during training (but then again, maybe it must in order for the network to do better, and there is no reason why it can’t grow at least somewhat). So… I’m not sure what the result of this experiment really means. A better experiment would have compared different initialization schemes. Some of the writing in this section is also wishy-washy: For example, saying that “the effective receptive field is smaller than the theoretical receptive field” seems vacuous since this must always be true… the question is how much smaller and the current experimental results do not quantify the relationship in a clear way.

Confidence in this Review

3-Expert (read the paper in detail, know the area, quite certain of my opinion)


Reviewer 3

Summary

The authors study the spatial extent of the effect of inputs on an output of deep convolutional networks, termed receptive field size. For most of the paper they do analysis on networks with either uniform or random weights and no nonlinearities and find a gaussian distribution that shrinks as square root of depth relative to maximum extent of the receptive field. They make few comments on nonlinear case and then they verify results experimentally. Then they look at the receptive field of common vision architecture before and after training and find that training increases receptive field size. The theoretical analysis is nicely done and good to have even though it is not clear to what extent it might be useful for trained architectures with nonlinearities, but that will be seen in the future (one candidate could be for better initialization).

Qualitative Assessment

While the average of receptive fields is gaussian (Figure on page 6), each individual filter seems to have a lot of shapes even in random weight case (Figure 1). Can you characterize these shapes somehow - even through numerical analysis? “This region in the input is the receptive field for that unit. This property of convolutional architectures makes CNNs spatially invariant, which is ideal for visual recognition tasks” - this is not what makes them spatially invariant Fig on page 6 - we see a blob that could be gaussian or other symmetric blob like distribution - does gaussian fit it well? 2.3: It would be good to write in one sentence what this paragraph is going to be about (what are you going to show). 4.2: Shouldn’t non-res net be tried - it is more basic.

Confidence in this Review

2-Confident (read it all; understood it all reasonably well)


Reviewer 4

Summary

The authors study the spread of gradient information in deep convolutional neural networks. In particular, the authors reason by a Fourier analysis that an element of the gradient at the loss layer of a convnet has a localized effect on elements of the gradient at hidden layers in a distribution they call the "receptive field", and that this localization has a normal distribution. The authors also present empirical evidence that the gradient receptive field of convnets looks Gaussian (or at least has diminishing support), and that the actual receptive field is smaller than an upper bound based on number of layers and kernel size.

Qualitative Assessment

I think this is a very interesting look at how information propagates through convnets. In particular, the fact that the gaussian-ness falls out of the combinatorics of the Fourier domain representation of convnets is a very neat trick which will likely inspire future work. The authors also considered a number of modern neural network innovations and their effects on the gradient receptive field, which was nice. Some of the math is a little bit hand-wavey but that is excusable as it was more or less clear what was going on. A suggestion is to more clearly state that $u(t)$ is the gradient at the network output and $o(t)$ is the gradient at the network input, and also that $t$ here is a 1D spatial dimension introduced for simplicity (perhaps choose another letter?). It seems that your experimental results use the same setup for $u(t)$ as in your theory, that is, $u(t)$ is an impulse centered at $t=0$. This is fine, but I also wonder what the edge effects look like, say if you had $u(t)=\delta(t-15)$ in the case of CIFAR-10. One criticism is that the experimental results are of a more qualitative nature, which by itself is not bad (e.g. Figure 1 is good for building intuition). Mostly I am kind of curious as to just how (non-)Gaussian some of the computed receptive fields actually are in a quantitative sense, rather than just looking at pictures. Also, while you considered the effect of architectural depth in Figure 1 to confirm your theoretical results, what would have been very interesting is training architectures of varying depth on CIFAR-10 or perhaps ImageNet, and then observing and comparing the shape, size, and distribution of the resulting receptive fields. Some errata: - In equation (10), left hand side should be the products not convolutions of U and V? - You might benefit from checking the paper for typos.

Confidence in this Review

2-Confident (read it all; understood it all reasonably well)


Reviewer 5

Summary

This paper studies the characteristics of receptive fields in deep convolutional networks. The investigation could be meaningful in that the research findings could potentially benefit research in multiple areas that make use of CNNs. The authors try to investigate the effects of sub-sampling, skip connections, dropout, and nonlinear activations. They show that the distribution of impact in a receptive field distributes as a Gaussian and suggest a few ways to potentially increase the effective receptive fields.

Qualitative Assessment

The authors try to investigate the effects of sub-sampling, skip connections, dropout, and nonlinear activations. They show that the distribution of impact in a receptive field distributes as a Gaussian and suggest a few ways to potentially increase the effective receptive fields. While the investigation seems to be meaningful, I found the conclusions to be a little bland and confused. There are a few places that I wasn't sure how the conclusion was derived. For example in section 2.5, it was stated "we can apply the same theory we developed above to understand networks with subsampling layers. However, with exponentially growing receptive field introduced by the subsampling or exponentially dilated convolutions, many more layers are needed to see the Gaussian shape clearly." It seems unclear how the same theory was applied to subsampling layers, and the second sentence didn't really draw any firm conclusion. Section 4.2 appears to be the only empirical evaluation on the assumptions. I feel the descriptions are somewhat vague and lack details. It is somewhat difficult to understand what they try to justify using the proposed experiments and what conclusions we could draw from that. There is a missing reference in section 2.6. "[] make extensive use of skip connections."

Confidence in this Review

2-Confident (read it all; understood it all reasonably well)


Reviewer 6

Summary

The paper attempts to “debug” deep convolutional neural networks (CNNs) by analyzing their receptive field in the visual tasks. It provides theoretical evidence that under certain assumptions the effective receptive field (ERF) is asymptotically distributed as a Gaussian and, therefore, smaller than the theoretical receptive field (i.e., smaller than expected). The authors discuss how different blocks of CNNs affect the ERF. They suggest that performance of CNNs can be improved if the ERF is increased or grows faster width depth. Most of their theoretical findings are supported in the experimental part on image datasets (mainly CIFAR-10).

Qualitative Assessment

Technical quality. - The quality of experiments is overall good, but still could be improved. For instance, it is not informative to provide absolute values of classification accuracy (e.g., accuracy of 89% on CIFAR-10 is neither good for today standards, nor meaningful). Instead, a comparison should be provided in which contribution of authors ideas are clear. Since there is such comparison (although, only in the rebuttal) for training speed clearly showing improvements, and questions of other reviewers regarding technical quality are addressed, I am able to give higher scores. The author should check if they use correlation or convolution in eq. (7)-(9) and (14). It should not be important for the distribution, but still seems that you use correlation while say that it's convolution. Novelty/originality. The idea to analyze the effective receptive field sounds novel. It would be interesting to see analysis of more nonlinearities inherent to networks. Potential impact or usefulness. This work potentially can be really useful and employed to improve initialization or architectures of CNNs. The authors provide some empirical evidence that making the effective receptive field larger by a special initialization method speeds up training of CNNs, which is quite valuable. The theoretical part can also be picked up and further extended by others. Clarity and presentation. - Additional proofreading would be nice. The paper has several rather minor errors/typos. Next, all headings should be lower case (e.g., Section 4.2) according to the NIPS style. All figures should have numbers and captions, otherwise, it can be hard to refer to them. Sentences containing equations should follow punctuation rules (e.g., equations through lines 78-82 and other instances). In several cases, additional text should be provided along with notations to ease reading of the paper. For instance, “...the receptive field size as n grows.” (line 240). So, what is ‘n’? Even though somewhere in the paper ‘n’ was denoted as something, the reader will have to search for it. - Also, few minor comments regarding references. 1) Stay consistent with proceeding titles, e.g., either use full titles as “Advances in neural information processing systems” or abbreviated as “NIPS” for all references. 2) If the paper from arXiv you refer to was published in another (peer reviewed) place, it’s better to cite that place. For example, “Deep Residual Learning for Image Recognition” was published at CVPR.

Confidence in this Review

1-Less confident (might not have understood significant parts)